# HDV Pathogenesis: Unravelling Ariadne’s Thread

**DOI:** 10.3390/v13050778

**Published:** 2021-04-28

**Authors:** Eirini D. Tseligka, Sophie Clément, Francesco Negro

**Affiliations:** 1Department of Pathology and Immunology, University of Geneva Medical School, 1211 Geneva 4, Switzerland; 2Division of Clinical Pathology, University Hospitals, 1211 Geneva 4, Switzerland; sophie.clement@unige.ch (S.C.); francesco.negro@hcuge.ch (F.N.); 3Division of Gastroenterology and Hepatology, University Hospitals, 1211 Geneva 4, Switzerland

**Keywords:** hepatitis Delta virus, HDV, hepatitis B virus HBV, HDV pathogenesis, co-infection, super-infection, acute HDV infection, chronic HDV infection

## Abstract

Hepatitis Delta virus (HDV) lies in between satellite viruses and viroids, as its unique molecular characteristics and life cycle cannot categorize it according to the standard taxonomy norms for viruses. Being a satellite virus of hepatitis B virus (HBV), HDV requires HBV envelope glycoproteins for its infection cycle and its transmission. HDV pathogenesis varies and depends on the mode of HDV and HBV infection; a simultaneous HDV and HBV infection will lead to an acute hepatitis that will resolve spontaneously in the majority of patients, whereas an HDV super-infection of a chronic HBV carrier will mainly result in the establishment of a chronic HDV infection that may progress towards cirrhosis, liver decompensation, and hepatocellular carcinoma (HCC). With this review, we aim to unravel Ariadne’s thread into the labyrinth of acute and chronic HDV infection pathogenesis and will provide insights into the complexity of this exciting topic by detailing the different players and mechanisms that shape the clinical outcome.

## 1. Introduction

Hepatitis Delta virus (HDV) is a defective virus and an obligate satellite of hepatitis B virus (HBV), necessitating its helper virus envelope proteins (HBsAg) to form its viral particles. Humans are the only natural reservoir of HDV, but other mammalian hosts have been identified capable of supporting the HDV life cycle, including chimpanzees and tree shrews (with the HBV as a helper virus), as well as woodchucks (with the woodchuck hepatitis virus as helper virus) (reviewed in [1]). While its genome replication and ribonucleoprotein (RNP) formation are independent of HBV, HDV egress requires HBV envelope proteins [2,3].

Despite forty years of epidemiological studies, the global number of HDV patients still remains elusive. The widespread implementation of HBV vaccination in high-income countries has limited the number of HBsAg susceptible persons, and subsequently, the number of HDV patients. In these countries, the epidemiology of HDV is dual, with an ageing cohort of local patients with advanced liver fibrosis and a younger generation from endemic countries that represents the incoming new infections [4]. Based on the literature published during the decade of 1980–1990, a recent meta-analysis estimated that the global number of HDV patients should reach 12 million in 2020 [5]. However, selection biases, technical limitations, inadequate screenings for anti-HDV in HBsAg-positive patients that represent the sole reliable source of HDV infection [6], and lack of HDV RNA testing to confirm ongoing viral replication can flaw the appraisal of the prevalence and health burden of HDV (reviewed in [4]). Contrary to HBV, HDV transmission from mother to offspring, and among homosexuals, has been rarely reported [6]. Therefore, epidemiological data of HDV prevalence and disease progression are not available in these groups and especially in babies, where the infection of HBV at birth will lead to chronic hepatitis B development. Of note, in Europe, up to 40% of anti-HDV-positive persons do not harbour detectable HDV replication [7].

The available antiviral therapies against chronic hepatitis D (CHD) lies on the administration of the pegylated (PEG) form of IFN-α with, however, low response rate and high frequency of adverse effects (reviewed in [1]). The administration of an entry inhibitor (Bulevirtide) has recently been the object of a conditional authorization by the European Medicines Agency (but not by other regulatory authorities) for chronically HDV infected patients with compensated liver disease [8]. The future role of combination therapies is still elusive and requires further investigation, as the therapeutic management of HDV remains unsatisfactory [4].

HDV pathogenesis can vary from asymptomatic cases to acute liver failure and CHD, which is associated with a faster progression towards cirrhosis, liver decompensation, and hepatocellular carcinoma (HCC) compared to HBV monoinfection. CHD is considered as the most aggressive form of chronic viral hepatitis posing major clinical challenges, especially in high endemic countries [9]. In the following chapter, we will describe the three different modalities of HDV infection and their respective pathogenesis.

## 2. The Three Moirai of HDV Infection and Their Clinical Outcome: From HDV-HBV Co- or Super-Infection to HDV Mono-Infection

Given its obligatory dependence on HBV to sustain its life cycle, HDV infection occurs via two modalities, i.e., as a simultaneous infection or co-infection with both HBV and HDV, or as an HDV super-infection of a chronic HBV carrier (Figure 1). The third modality of HDV infection refers to HDV monoinfection of the hepatocytes in the absence of HBV infection, which is non-productive but can be rescued by the helper virus at a later stage. All the pathologic changes of HDV infection are limited to the liver, as the virus requires the expression of the sodium taurocholate co-transporting polypeptide (NTCP) entry receptor, which is located at the basolateral membrane of differentiated hepatocytes [10].

### 2.1. HDV-HBV Co-Infection

Co-infection of HDV and HBV leads to acute hepatitis, whose symptoms are indistinguishable from a typical acute hepatitis B and can range from mild to severe and even acute liver failure, which is characterized by a massive hepatocyte necrosis. A study from Yurdaydin and colleagues demonstrated that 17% of the HBV-HDV co-infected patients developed severe hepatitis [11]. A minority of co-infected patients may progress even to acute liver failure [11,12]. Without liver transplantation, the mortality of these patients may reach 80% [13]. However, when patients with acute liver failure associated with HBV monoinfection are age-matched to patients with HBV/HDV co-infection, the difference in mortality cannot be observed anymore [14].

Acute hepatitis occurs after an incubation period of 3–7 weeks with a preicteric and icteric phase that is not always observed [15]. During the preicteric phase, several nonspecific symptoms including fatigue, lethargy, anorexia or nausea, and biochemical markers, such as elevated serum ALT and AST levels are observed. Acute HDV-HBV co-infection can present either with a single peak, or with two distinct peaks, of liver enzyme elevations, mostly separated by 2–5 weeks [16] (Figure 2). This re-increase of serum ALT and AST levels, also called biphasic hepatitis, emerges after a first period of improvement and is thought to be caused by sequential spreading of HBV followed by HDV [17]. Typically, HBV-related peak is associated with markers of primary HBV infection, e.g., HBsAg, HBV DNA and anti-HBc IgM, and precedes HDV-related peak, where increasing titers of anti-HDV IgM and IgG are detected. An inverse pattern has been reported, especially in animal models [18], and attributed to the relative titers of HBV and HDV infection.

The majority of the co-infected and immunocompetent patients (90–95%) will resolve both viral infections. HDV RNA, an early and sensitive marker of HDV replication in acute HDV infection [19], and the presence of anti-HDV IgM and IgG will disappear during early convalescence [6]. HDV diagnosis relies almost exclusively in the detection of total levels of anti-HDV antibodies, and is subsequently confirmed by the detection of HDV RNA by polymerase chain reaction (PCR) [20]. Serum HDV antigen (HDAg) appears early upon HDV-HBV co-infection, but it disappears quickly, therefore requiring repeated testing [21]. Serum HDAg is not a suitable tool for HDV diagnosis, as it cannot be directly detected by enzyme immunoassay or radioimmunoassay, due to antigen sequestration in immune complexes with high tittered circulating antibodies [22], and requires the application of immunoblot assay under denaturating conditions, a technique which is difficult to apply for routine detection [22]. The appearance of anti-HBs antibodies will indicate HBV viral clearance (Figure 2).

A small proportion of HDV-HBV co-infected patients (less than 5%) will progress to chronic infection [16] with the risk of developing liver cirrhosis and HCC (Figure 1). Explicit information about CHD and its clinical outcomes are detailed in the following section.

### 2.2. HDV-HBV Super-Infection

Alternatively, HDV can super-infect a chronic HBV carrier and progress to an acute or a chronic HDV infection [23] with various clinical outcomes, including patients with unspecific symptoms [15]. The pre-existing HBV infection provides the ultimate virologic background for a rapid HDV spread that will exacerbate the pre-existing liver disease [24]. Acute hepatitis D due to super-infection can be mistakenly considered either as a hepatitis B reactivation or an acute HBV hepatitis in a previously non-diagnosed HBsAg carrier [11,25]. In HDV super-infected patients, the risk of severe hepatitis is higher than in HBV mono-infected patients [12], with frequent evolution towards an acute liver failure requiring liver transplantation. The original multicentre report from 7 European centres included 532 patients with acute benign hepatitis B from Italy, 111 patients with acute liver failure from Italy, France and England, and 18 patients with acute hepatitis D that lacked anti-HBc IgM [12]. The investigators suggested that, in these patients, the associated liver failure was most likely associated with an HDV super-infection event [12]. In an Italian series, 70% of patients with CHD recalled an episode of acute hepatitis in their history [26].

In the course of HDV super-infection, HBsAg positivity precedes the detection of HDV RNA, due to the already pre-established HBV chronic infection characterized by the absence of anti-HBc IgM antibodies (Figure 3). Upon the appearance of HDV RNA, HBsAg levels will fluctuate in patients’ serum. During the acute phase of HDV super-infection, HBV DNA level decreases to rebound after HDV clearance, although this is not always systematic [27] (Figure 3A). The resolution of HDV infection coincides with the decline of anti-HDV IgM and IgG immunoglobulins, and the normalization of ALT levels (Figure 3A).

While a low rate (less than 5–10%) of HBsAg super-infected carriers may undergo a self-limited hepatitis leading to HBV clearance [28,29] (Figure 1), the vast majority of HDV super-infected patients (more than 90%) will progress to chronic infection with both HBV and HDV viruses [11,13,16]. Active replication of HBV and/or HDV remains a threatening factor towards liver decompensation during chronic HDV infection [30]. The evolution towards chronic infection is associated with the persistence of HDV RNA and IgM and IgG anti-HDV (Figure 3B) [15]. ALT and AST levels are persistently elevated in most of the patients [15]. IgM antibodies are also detectable in primary infection and persist with progression to chronicity, although they may be absent in some patients from Africa [31].

CHD is considered the most severe form of chronic viral hepatitis, burdened with major clinical outcomes [32]. Like all immune-mediated disorders, a wide variability in the severity has been reported. A particularly aggressive course was observed among intravenous drug users since early studies [26]. The rapid spread from one to the next susceptible patient may have favoured the selection of more virulent strains, similarly to what was suggested by a chimpanzee study, where the severity of serially passaged HDV was associated with shortened incubation time and increased severity of acute hepatitis, irrespective of HDV replication levels [33]. Histological analysis demonstrates a severe hepatitis with advanced fibrosis in super-infected HDV patients [26,34,35], that undergo accelerated progression to cirrhosis [26,36,37,38], an increased risk of hepatic decompensation, and HCC leading to death [9,13,39], when compared to HBV monoinfected patients. Progression towards cirrhosis can be rapid [26,38] but also indolent [23]. In an early study of 75 patients with chronic hepatitis (HBsAg carriers with intrahepatic delta antigen) at enrolment, 39% developed cirrhosis or liver failure after a follow up (FU) of up to 6 years [26]. The same cohort was later followed for an average of 12 years [40], and the proportion of cirrhosis increased with the duration of follow-up, i.e., 23%, 41%, and 77% in the first, second, and third decade of disease, respectively. The progression to cirrhosis after an average 20-year follow-up was somehow less (42%) in another more recent, retrospective, large series from Italy [39]. Overall, the risk of developing cirrhosis in patients co-infected with HBV and HDV seems two-fold higher compared to patients infected with HBV alone [41]. In a small series from Turin [30], 85 chronic HBsAg and anti-HDV carriers were followed for an average time of 10 years. Unfavourable clinical outcomes (HCC, ascites, or liver failure) occurred in 77% of the patients with detectable levels of both HDV RNA and HBV DNA in their serum, whereas 21% of patients with HDV RNA and undetectable HBV RNA were diagnosed with ascites [30]. These results were echoed by a larger study from Taiwan, where the simultaneous presence of serum HBV DNA and HDV RNA was associated with a lower remission rate vs. those negative for both viruses (21.7% vs. 69.2%; *p* < 0.001), whereas the difference was only numerically lower than that observed among patients positive only for HBV DNA (26.4%) or HDV RNA (24.3%). However, the cumulative survival rate at 15 years was 57.6% among patients with HBV DNA vs. 78.3% among HBV negative patients [42].

The impact of HDV infection on the rate of HCC development in HBV-positive patients has been a subject of controversy, as despite the high rate of progression to cirrhosis, not all the studies demonstrate an increased rate of HCC [30]. Some studies suggest that the major complication of CHD is decompensated cirrhosis, rather than HCC [43,44], implying that liver failure and liver-related death precede HCC development. In a landmark retrospective European study enrolling 200 patients with compensated cirrhosis [9], the presence of anti-HDV antibodies induced a 3-fold increase of HCC and a two-fold increase of mortality compared to HBV monoinfection. Several adjustments and stratification, according to the presence of anti-HDV and HBeAg, highlighted that the patients who were anti-HDV positive/HBeAg negative had an estimated 5-year risk of HCC of 13%, compared to 4 and 2% among anti-HDV negative/HBeAg negative and anti-HDV negative/HBeAg positive patients, respectively. No difference in terms of mortality was reported. In a large study from the Swiss HIV Cohort, where HBV replication was presumably suppressed by antiretrovirals, HDV infection was independently associated with mortality and liver-related events, including HCC [7]. HDV replication is a major determinant of HCC development relatively to HBV [45,46]. A recent systematic review of the literature and meta-analysis of the available data from our group highlighted an association of CHD with an increased risk of developing HCC, compared to HBV monoinfection [47]. This analysis of 93 studies, despite an important study heterogeneity, showed a significantly increased risk of HCC in patients with CHD, with pooled OR of 1.28; 95% CI 1.05–1.57; I^2^ = 67.0%. The association was stronger considering only prospective cohort studies (pooled OR 2.77; 95% CI 1.79–4.28), those with HIV-infected patients (pooled OR 7.13; 95% CI 2.83–17.92) where heterogeneity was less, and in general, in studies with well-defined inclusion criteria and adjustments for confounders, hinting at the importance of a robust study design. This was also evident considering that the strength of the association decreased, or became insignificant, in studies with high risk of bias, or in studies carried out before 2010. Regarding the geographical origin of patients, interestingly, the association was confirmed in Asian studies, but not in studies originating elsewhere. Lack of data prevented the analysis of the respective contribution of HBV and HDV genotypes.

### 2.3. HDV Replication in the Absence of HBV

The third modality of HDV infection refers to the monoinfection of susceptible hepatocytes by HDV in the absence of a helper Hepadnavirus, and has been the apple of discord over the years for its potential impact on liver transplantation. In this case, a “helper independent HDV infection” or “latent” HDV infection has been suggested as markers of HDV replication have been identified in the liver and serum of the patients in the absence of HBV markers [48]. In an early series of patients undergoing liver transplantation and receiving robust anti-HBV immunoprophylaxis [48], HDV infection of the grafted liver recurred early without signs of liver damage or HBV reactivation. As soon as HBV recurred, hepatitis flared associated with serological markers of both HDV and HBV. A similar finding was reported in at least one patient transplanted in a series from Paris [49]. In an attempt to reproduce this model in susceptible animals, woodchucks never exposed to the WHV were infected with serum from an acutely infected animal, where the HDV titer was about 1700-fold higher that of WHV. This led to the expression of HDAg in scattered (~1%) hepatocytes in the absence of a productive infection, implying a very infrequent, if not absent, co-infection of hepatocytes by HDV and WHV [50]. A second challenge, this time with WHV, resulted in a productive HDV infection up to 33 days after HDV monoinfection. The persistence of HDV in the absence of HBV was also tested in the chimpanzee [51], but here, among the two animals first inoculated with HDV alone, only the one exposed to HBV one week later developed a dual HDV/HBV infection, whereas HDV infection could not be rescued when the HBV was inoculated 4 weeks later. The same authors analysed, in detail, the dynamic of HDV infection early after liver transplantation in humans, showing that, using sensitive PCR-based assays for HDV and HBV, both viruses were invariably detected in serum, albeit with very low levels of HBV replication, following the early post-liver transplantation incubation period, suggesting that some degree of co-infection was occurring despite the absence of detectable HBsAg in serum due to the effective immunoprophylaxis. The above re-evaluation of viral titers ruled out the possibility of an isolated HDV infection [51]. Thus, these findings showed that HDV does not seem to undergo a bona fide latent infection early after liver transplantation. 

However, in a subsequent retrospective analysis of patients transplanted at Hannover [52], HDAg was detectable in transplanted livers of 6/26 patients in the complete absence of liver HBV DNA, cccDNA, serum HBsAg, and HDV RNA for up to 19 months from transplant. The intrahepatic HDV replication in the complete absence of HBV was also elegantly proven in a humanized mice model infected with cell culture-derived HDV particles or with a serum sample from an entecavir-treated HBV/HDV co-infected patient with undetectable HBV viremia [53]. This so-called latent HDV infection, occurring in a small minority of hepatocytes (~1.5%), was not associated with liver damage (assessed by detection of hepatocyte apoptosis by a TUNEL assay) but was rescued by a super-infection with HBV 3 and 6 weeks later. The question about whether this model of infection may have relevance in transplanted patients remains debated. 

The persistence of HDV in the liver for at least 6 weeks in the absence of HBV [53] raised the question of whether hepatic viruses, other than HBV, could trigger its propagation. A recent study from Vargas et al., [54] demonstrated the ability of HDV to use glycoproteins from other viruses, such as vesiculovirus, flavivirus and hepacivirus by applying an in vitro expression system, or by co-infecting the cells with hepatitis C virus (HCV) or dengue virus. The formation of infectious HDV particles with unconventional glycoproteins could occur as an efficient packaging of HDV RNPs and viral egress was reported in the extracellular medium of the co-infected cells. Those particles were able to efficiently enter into cells expressing the relevant receptors. Importantly enough, the researchers demonstrated that HDV propagation could be mediated by HCV in the liver of experimentally co-infected humanized mice for several months.

A recent analysis of serum from 160 HCV infected patients that were under treatment with a combination of PEG-IFN-α and ribavirin, revealed the presence of anti-HDV antibodies in the sera from two patients. Interestingly enough, these patients were negative for all serological or molecular markers of HBV infection, such as HBsAg, anti-HBc antibodies and HBV DNA, as assessed by two different types of PCR (qPCR and droplet digital PCR-ddPCR). This is the first study in patients that describes the presence of HDV infection, as shown by the detection of HDV antibodies, in chronically HCV infected patients without the evidence of ongoing or past HBV infection [55]. The above studies suggest the in vivo ability of HCV to act as a helper virus of HDV in the absence of HBV. Additional studies including larger cohorts, with no previous HBV exposure, are required to explicitly investigate the aforementioned observations.

## 3. Mechanisms of HDV Induced Pathogenesis

HDV hepatotropism limits its pathologic changes to the liver with the clinical sequelae to be extremely variable, ranging from acute liver failure to asymptomatic carrier state and chronic infection. The exact mechanisms of HDV induced liver damage are still barely identified, and the severity of the clinical course is influenced by several factors. Some studies suggest that HDV can cause direct cytopathic damage during acute infection, whereas immune-mediated damage predominates during chronic infection [56]. Other studies claim that HDV, like other hepatic viruses, is not directly cytopathic to infected hepatocytes, and it is mostly the innate and adaptive immune responses that contribute to the liver immunopathogenesis [57].

HDV pathogenesis is orchestrated by a breadth of different factors including HDV and/or HBV and/or host-associated factors (Figure 4). Some of these factors possess a clear causality effect with HDV pathogenesis, whereas, for others, a more correlative effect is proposed, therefore necessitating additional investigations. In the following sections, we will discuss in detail their role on HDV pathogenesis.

### 3.1. HDV Associated Factors

#### 3.1.1. HDV Genotype

The role of HDV genotypes has been associated with HDV pathogenesis as specific clinical features seem to cluster in different geographical areas [58,59,60]. However, since the 8 genotypes of HDV are largely distributed in well-defined areas of the world, it is difficult to disentangle the effect due to genotypes from that potentially linked to ethnicity.

Most natural history studies have been carried out in the Western world, where genotype 1 prevails [61,62] and has been associated with a more severe course of HDV pathogenesis. Genotype 1 is also predominant in low-to middle-income countries, such as Mongolia, which is highly endemic for HDV infection [63,64]. Genotype 2 is prevalent in the Far East, where acute liver failure due to HDV seems less frequent, and progression to end-stage liver disease appears slower [42,59,65,66], although a genotype 2 variant isolated from Japanese patients (the Miyako strain) [67] has been reported to be associated with a faster progression towards cirrhosis. Genotype 3 is mostly found in South America and has been associated with a severe form of hepatitis. Here, acute liver failure has been, consistently and frequently, reported in severe outbreaks of acute hepatitis D affecting isolated communities of the Yukpa people of Venezuela [68,69], the Sierra Nevada de Santa Marta in Colombia [70], and the Western Amazon basin [58,71].

Five additional HDV genotypes have been described upon the classification of the previously assigned genotype 2b as genotype 4, and the classification of African sequences into the genotypes 5 to 8 [62,72]. The disease features of these genotypes remain poorly characterized so far.

In their recent paper, Le Gal et al. highlighted the classification of HDV genotypes to one or four subgenotypes per genotype according to their intersubgenotype similarity over the whole genome sequence [73]. HDV-1a and HDV-1b subgenotypes were restricted to Africa and Madagascar, HDV-1c to the Oceania islands and HDV-1d to the Middle East, eastern and western Europe, Asia, and North America. HDV-2a circulated to Taiwan and Japan, and HDV-2b to Siberia. HDV-4a and HDV-4b circulated in the Far East, and HDV-7a and HDV-7b were detected in Cameroonian patients. HDV-5 and HDV-8 genotypes segregated into at least two subgenotypes without well-defined geographical specificity and no distinct information about HDV-3 subgenotypes are available so far [73]. Further studies are required to identify the association between HDV subgenotypes and viral pathogenesis.

Mixed infection with different genotypes, and even super-infection, have been reported, especially in patients with high risk for multiple exposures to HDV. In this case, there is a sole dominant genotype, whereas the minor one represents approximately 10 percent of the total viral population [74].

#### 3.1.2. HDAg Expression

HDAg expression coincides with HDV replication, is quite robust within an infected hepatocyte, and was demonstrated to be barely influenced by mutations that have been identified in several HDV isolates. In 1986, the CAR HDV isolate was linked with an acute liver failure outbreak in the region of Central African Republic (CAR) [75]. The CAR HDV isolate, which was successfully transmitted and isolated from the liver of woodchuck upon their inoculation with sera from patients with acute liver failure, showed a highly specific liver pathogenicity, including spongiocytic hepatitis in both woodchucks and humans. This isolate bared a mutation (T to A) at nucleotide 1013 of the antigenomic RNA, which converted the amber stop codon (TAG) to a codon for lysine (AAG) that produced a unique HDAg species of 28kDa size [76], which was identified in the livers and sera of the infected hosts [77]. Of note, the reference HDV isolate bears a U to C modification on the genomic RNA at nucleotide 1012 and will give rise to the small (24kDa) and large (27kDa) HDAg [78]. Sequencing analysis of the CAR HDV isolate and its comparison with the American, Japanese, Taiwanese, French, Italian, and Nauru HDV isolates, revealed a variability of 1.7 to 21.5% at the nucleic acid level and of 1.9 to 28.7% at the amino acid level [76]. CAR HDV isolate was most closely related to the Italian HDV isolate, demonstrated an extremely low genome replication, and its HDAg shared common biological functions with the prototype L-HDAg, including the replication inhibition of HDV RNA [79]. There is no additional information about the prevalence of this isolate and no recent data are available so far.

The liver pathology hallmark, based on initial animal studies, is an eosinophilic degeneration of hepatocytes [80]. Additionally, early in vitro studies suggested a direct cytopathic effect of HDV [81,82]. However, these observations failed to be reproduced in transgenic mice expressing either L-HDAg or S-HDAg, therefore suggesting no association between HDAg expression and liver damage [83].

No correlation between HDV replication levels and any histological feature was identified in an international study with 80 HDV chronically infected patients at different stages of fibrosis [84]. However, in the same study, Zachou et al. demonstrated a weak correlation between the serum levels of HBsAg and the histological activity of the liver. In another clinical study, Negro et al. demonstrated a significant positive correlation between the number of HDAg-positive cells and the extent of portal inflammation, upon analysis of the intrahepatic expression of HDAg, the morphologic features of HDV hepatitis, and the outcome of liver disease in 101 patients. HDV elimination reduced the degree of inflammation and intrahepatic HDAg expression, while the initial morphologic lesion of the liver did not impact on HDV disease. The results of this study suggested that the immune response plays a crucial role in the pathogenesis of HDV hepatitis [40].

#### 3.1.3. HDV Persistent Replication

Persistent HDV replication, as assessed by HDV RNA detection in the serum, has a major impact on the progression towards CHD and was suggested to be the only predictor of liver related mortality [39]. HDV RNA levels were independently associated with progression to cirrhosis (OR = 1.60, 95% CI 1.20–2.12, *p* = 0.007) and development of HCC (OR = 1.88, 95% CI 1.11–3.19, *p* = 0.019) in a FU study on 105 non-cirrhotic patients [45]. These results are consistent with early observations from the Turin cohort, where, despite the initial rapid progression to cirrhosis, some patients may experience a reduction of HDV replication with time, which is associated with a pattern of inactive cirrhosis, characterized by reduced necroinflammation, stable for years [40,66,85]. A relative stability—even at the cirrhotic stage—has been also reported in children [86]. Roulot et al. recently highlighted that persistent HDV replication was one of the critical determinants of HDV severity, as it was associated with significantly increased risk of hepatic complications, including liver decompensation, HCC, and death in a large French cohort [87].

### 3.2. Host-Associated Factors

#### 3.2.1. Interaction with the Cell Machinery

Previous studies demonstrated an interaction of HDV RNA, and its proteins S and L-HDAg, with the cell machinery [88,89]. Only a part of these interactions has been elucidated. HDV proteins interact with numerous cell factors that are involved in HDV transcription, replication and even pathogenesis [3]. A cell proteome analysis in human embryonic kidney HEK-293 identified that HDV replication altered the expression of 89 out of 3000 proteins that were quantified [90]. The majority of these proteins was associated with pyruvate metabolism (Figure 5C1) and cell cycle regulation (Figure 5A). The expression of p53 was downregulated, and the G2/M DNA damage checkpoint was highly affected, therefore implicating a potential role of these pathways in HDV associated HCC. Additional experimentations demonstrated that HDV can induce cell cycle arrest [91], cell death [81], and impair cell proliferation [92].

HDV was shown to increase histone H3 acetylation within the clusterin promoter that enhanced its expression (Figure 5B) [93]. The above epigenetic regulation along with the fact that clusterin was overexpressed in 89% of human HCC cases [94], strongly suggests that HDV can induce carcinogenesis.

HDV can also promote oxidative stress in the endoplasmic reticulum (ER) of hepatic cell lines by interacting with the NADPH oxidase (Nox) family (Figure 5C2). L-HDAg interacts with NOX-4 inducing the release of reactive oxygen species (ROS) which in turn activate the signal transducer and activator of transcription-3 (STAT-3) and the nuclear factor kappa B (NF-κB) pathway (Figure 5C2) [95]. This effect was dramatically decreased in the presence of antioxidants or calcium inhibitors.

HDV may induce intrahepatic inflammation via the activation of NF-kB signalling. In vitro studies revealed that only L-HDAg can activate the tumour necrosis factor alpha (TNF-α) induced NF-κB signalling, despite the fact that both HDAg isoforms can interact with the TNF receptor-associated factor 2 (TRAF2). NF-κB activation by L-HDAg is independent of L-HDAg farnesylation, is mediated through the death receptor TNFR1 (tumour necrosis factor receptor 1) signalling cascade and induces the expression levels of Cyclooxygenase-2 (COX-2) (Figure 5C4) [96]. L-HDAg farnesylation is a post-translational modification, where a farnesyl lipid group (C211XXQ box) is covalently linked by a cellular farnesyltransferase to the cysteine at position 211 of L-HDAg [97]. ER stress and NF-κB activation was shown to be activated by the translocation of L-HDAg to the ER, during the viral assembly with the residing HBV glycoproteins [98].

L-HDAg interacts with various signalling pathways implicated in epithelial to mesenchymal transition (EMT) [99], wound healing, and fibrosis [100]. The transforming growth factor β (TGF-β) pathway is involved in liver regeneration and in the fibrotic to cirrhotic transformation upon viral infection. Choi et al. demonstrated that only L-HDAg, induced the signal cascades of TGF-β, c-Jun-induced pathway and enhanced the protein expression level of TGF-β–induced plasminogen activator inhibitor-1 (Figure 5C3) [100]. L-HDAg farnesylation, had a critical role in the activation of these signaling cascades. Additionally, the synergistic interaction of L-HDAg and HBx (HBV X protein) activated TGF-β and the activator protein-1 (AP-1) pathway of transcription factors by binding to Smad3, STAT3 and inducing c-Jun pathway (Figure 4) [100]. Therefore, TGF-β signalling regulation via L-HDAg farnesylation might represent an alternative mechanism of HDV pathogenesis [100].

Goto et al. showed that only L-HDAg could activate the serum response factor (SRF)-associated transcription pathway, as opposed to the small isoform S-HDAg (Figure 5) [101]. SRF is a transcription factor that binds to the serum response element (SRE) and mediates serum and growth factor induced transcription from the c-Fos proto-oncogene (Figure 5C4) [102]. The same group later demonstrated the synergistic activation of the SRE-dependent pathway upon the interaction of L-HDAg with the HBx protein [103]. Further experimental studies are required to elucidate the implication of this pathway on HDV pathogenesis.

The expression of glutathione S-transferase P1 (GSTP1), a tumour suppressor gene, is typically downregulated in liver samples from patients infected with hepatotropic viruses. The inhibition of GSTP1 expression has been well studied in HCC tumorigenesis [104]. S-HDAg was shown to bind directly and downregulate GSTP1 in a human foetal hepatocyte cell line in vitro. This led to accumulation of cellular ROS, induction of apoptosis and selective pressure for malignant transformation (Figure 5C5) [105]. Therefore, the inhibition of GSTP1 expression by S-HDAg represents a novel potential mechanism of HDV pathogenesis.

#### 3.2.2. Innate Immune Response

Immune-mediated liver damage upon HDV infection is an important factor of HDV pathogenesis. A strong or a dysregulated innate immune response to a viral infection can lead to severe immunopathogenesis [106]. Up to date, the factors that mediate the interaction between HDV and host innate immune system have not been explicitly characterized. While HBV has been classically considered to be poorly recognized by the innate immune system [107], HDV can induce the expression of interferon (IFN)-β and IFN-λ, of pro-inflammatory cytokines and interferon stimulated genes (ISGs) in vitro [108,109,110,111] and in vivo [112,113] (Figure 6). IFN-λs or type III interferons are endogenous antiviral cytokines, functionally similar to type I interferons, that activate the intracellular JAK-STAT pathway and regulate the transcription of ISGs. In vitro, L-HDAg was shown to inhibit HBV replication by trans-activating the IFN-inducible *MxA* (myxovirus resistance protein A) gene [109] and HDV RNA accumulation induced a strong type I IFN response by stimulating the expression of *RSAD2* (Radical S-Adenosyl Methionine Domain Containing 2) and *MxA* genes [108]. Finally, the misfolded HDV ribozyme sequence was found to induce protein kinase R [110]. HDV is sensed by the RIG-I-like receptor (RLR) group and more precisely by MDA5 [114]. Zhang et al. demonstrated that active replication is required for innate sensing, without, however, pointing out the pathogen-associated molecular pattern (PAMP) of HDV which is recognised by the MDA5 [114].

Despite the intact innate immune response mediated by HDV, IFN-α failed to inhibit HDV in vitro [110]. A previous in vitro study demonstrated that HDV replication inhibited the tyrosine phosphorylation of JAK kinase Tyk2, thereby impairing the activation and translocation of STAT1 and STAT2 to the hepatocyte nucleus and inhibiting the antiviral cell response to IFN-α [115].

Natural killer (NK) cells are innate effector cells representing 30–40% of all intrahepatic lymphocytes [116,117] with a crucial role in antiviral defence against HBV and HCV infection [118,119,120]. Importantly enough, NK cells drive liver pathology in chronic HBV infection [121]. The role of NK cells in the peripheral blood of chronically infected HDV patients was recently investigated [122]. The researchers assessed the frequency and differentiation phenotype of NK cells prior to and upon IFN-α treatment in chronically infected HDV, or HBV patients and healthy controls. While untreated HDV patients showed an increased frequency of NK cells with unaltered phenotypic differentiation status, those that have been long-term treated with IFN-α demonstrated a loss of terminally differentiated NK cells and an enrichment in immature NK cell subsets (Figure 6). The NK cell differentiation profile has not been so far associated with HDV liver pathogenesis progression and therefore more studies are required to fully elucidate their role.

#### 3.2.3. Adaptive Immune Response

HDV adaptive immune response has been highlighted by the presence of specific T cell response to HDAg in the peripheral blood of chronically infected HDV patients. The presence of specific anti-HDV T cells is inversely correlated with HDV-induced liver disease activity [123]. A comparative study of 76 patients chronically infected with hepatitis B, C, or D viruses demonstrated that patients with CHD had the highest frequency of CD4^+^ cytotoxic T lymphocytes (*p* = 0.04 vs. HBV and HCV patients) [124]. These cytotoxic CD4^+^ T cells differ from the helper CD4^+^ T cells as they are perforin positive and share common features with CD8+ T cells, therefore being able to kill virus-infected cells (no information about the expression of granzymes in these cells was provided). The increased number of cytotoxic CD4^+^ T cells in patients with more advanced liver disease is considered as one important factor for the more severe course of viral hepatitis in the elderly (Figure 7) [124].

The role of CD8^+^ T cells in chronic HDV infection was only recently studied following the identification of an extensive set of CD8^+^ T cell epitopes [125,126]. HDV-specific CD8^+^ T cells against HDV peptides were detected upon the administration of an HDV DNA vaccine to mice and were also reported in two patients that cleared HDV infection. The above suggests that appropriate T cell response seems to be tightly correlated with HDV clearance [127].

In another study, Kefalakes et al. isolated peripheral blood mononuclear cells from 28 patients chronically infected with HDV and HBV and identified HDV specific CD8^+^ T cell epitopes, underlining the high immunogenicity of HDV, despite its small size and physicochemical properties (hydrophobic amino acid sequence) [126]. They also demonstrated that HDV specific CD8^+^ T cells were as frequent as HBV CD8^+^ T cells, less frequent than those of Epstein–Barr Virus, cytomegalovirus, or influenza virus and the least exhausted as they were not expressing the terminally differentiated CD57 marker [126]. This subset of activated HDV-specific CD8^+^ T cells could target HDV conserved epitopes and contribute to disease progression. On the other hand, the subset of memory-like HDV-specific CD8^+^ T cells was associated with escape variants with less human leukocyte antigen (HLA) binding and less T cell activation, therefore being unable to clear HDV infection (Figure 7). This study requires more investigations to unravel the exact role of HDV-specific CD8^+^ T cells, as the researchers did not study the role of CD8^+^ tissue-resident memory T cells and therefore conclusions must be drawn with caution.

The role of mucosa-associated invariant T (MAIT) cells on HDV infection has not been extensively studied so far [128]. MAIT cells are unique innate-like T cells that comprise up to 50% of the intrahepatic immune cell population (even in the healthy liver), secrete inflammatory cytokines (IFN-γ and TNF) and can act cytotoxically against infected cells. In the liver, MAIT cell function is regulated by interleukin-7 (IL-7), which is secreted from the hepatocytes under inflammatory conditions [129]. MAIT cells can promote mitogenic and proinflammatory functions of fibrogenic cells [130] and contribute to tissue remodelling [131]. In vivo, MAIT cell deficient mice were resistant to liver fibrosis, whereas MAIT cell enriched mice demonstrated increased liver fibrosis [130]. MAIT cell activation in the liver was also correlated with tumour growth, as they produced IL-17 that promoted macrophage differentiation into the inflammatory M2 subtype and stimulated the vascular endothelial growth factor (VEGF)-mediated angiogenesis (reviewed in [132]). In chronically infected HDV patients, the population of circulating and liver resident MAIT cells was dramatically decreased, as opposed to the population of MAIT cells in HBV monoinfected patients. The remaining population of MAIT cells exhibited a functionally impaired responsiveness in chronically infected HDV patients. Increased levels of the proinflammatory cytokines IL-12 and IL-18 that could promote MAIT cell death, as well as monocyte activation, were reported in HDV patients, suggesting that upon HDV infection, the normally abundant MAIT cells in the peripheral blood and the liver are activated, become functionally impaired, and at a later stage, are depleted (Figure 7) [128]. The above studies do not show any potential contribution of MAIT cells on HDV pathogenesis. However, additional studies highlighting the causality behind the severe depletion of this cell population during chronic HDV infection might unravel other properties of these cells that could be associated with HDV liver disease pathogenesis.

The adaptive immune response against chronically infected HDV patients seems to be insufficient to resolve HDV infection, therefore suggesting that the innate immunity may have a more critical role to limit HDV infection [133,134,135].

#### 3.2.4. Patients’ Risk Profile

The usual predictive factors that define the patients’ risk profile and their correlation with the adverse clinical outcome apply to CHD as well. Male sex [136], older age [137,138], and co-infection with HIV [139] have been reported to correlate with liver disease progression and clinical outcomes.

The presence of genetic polymorphisms in the gene of *IL28B* (Interleukin 28B) is also implicated in HDV infection and pathogenesis. *IL28B* gene encodes for IFN-λ3. Polymorphisms in the *IL28B* gene were initially associated with the spontaneous or treatment induced clearance of HCV [140,141,142,143]. Up to date, the SNP rs12979860 represents the strongest genetic association with any chronic viral infection and its treatment response. The same polymorphisms in the *IL28B* gene were reported for both HBV and/or HDV persistence [144].

### 3.3. Helper Virus Associated Factors

#### 3.3.1. HBV Genotype

HBV genotypes also appear to affect the clinical outcomes of CHD, although data is scarce. A prospective study analysed 194 dually infected patients from Taiwan (mostly infected with genotypes B and C of HBV) [42]. Although, at enrolment, there were no differences in the relative proportion of chronic hepatitis, cirrhosis, HCC, or remission cases among those with genotypes B vs. C, after a median FU of 135 months, HBV genotype C was associated with a significantly lower remission rate (0 vs. 32.1%), higher incidence of cirrhosis (34.8% vs. 8.9%) and adverse clinical outcomes (including cirrhosis, HCC, and mortality due to hepatic failure; 70.0% vs. 33.9%) than genotype B. By multivariate analysis, age, HBV genotype C (besides HDV genotype 1) were independent factors for unfavourable clinical outcome [42]. In a smaller, cross-sectional study from Brazil, HDV viral load was lower in HBV genotype A patients compared to patients with genotype D or F, although this difference did not match the clinical outcome, probably affected by the low number of HDV patients [145]. These data require further studies to confirm their clinical significance.

#### 3.3.2. HBV Replication

Active HBV replication has a critical role in deteriorating liver damage in patients with chronic HDV infection, an observation that is in line with HDV dependency on its helper virus [146]. However, the typical consequence upon HDV super-infection and the acute phase of HDV replication is the suppression of HBV DNA [24], that has been reported in both liver and serum of humans, animal, and in vitro models [146,147,148]. The mechanisms of HDV and HBV interference still remain poorly characterized, although in vitro studies demonstrated that S-HDAg strongly inhibits HBV mRNA synthesis or stability [148], while both isoforms of HDAg can activate the innate immune response (IFN-α inducible *MxA* gene) and repress HBV enhancers 1 and 2 [109]. In the above cases, with minimum HBV replication, the liver damage of chronically infected HDV patients, is mostly attributed to HDV rather than HBV. HBV replication suppression, might also be a cause of multiple co-infections with viruses other than HDV, that are present in the patients [3]. Interestingly, in a Chinese study, a late HBV DNA reactivation was reported, which, according to the researchers, may impact the progression to end-stage liver disease [149].

## 4. Conclusions

HDV, the satellite virus of HBV, with million carriers worldwide, remains a fascinating and unique virus with many unravelled steps of its life cycle. HDV infection can only be achieved in the presence of its helper virus, HBV. An HBV-HDV simultaneous infection (or co-infection) will, in the vast majority, be self-limited and lead to viral clearance, while an HDV super-infection of chronically infected HBV carriers will almost universally lead to chronic HDV infection. Chronic HDV infection is considered as the most severe form of viral hepatitis that is associated with faster progression towards end stage liver diseases, including cirrhosis, liver decompensation and HCC. More than one factors seem to be involved in the pathogenesis of HDV hepatitis and end stage liver diseases, implicating HDV, HBV, and host associated factors, underlining the complexity of this multifaceted disease. A more explicit and in-depth knowledge of the HDV life cycle will provide critical information to better understand the mechanisms of HDV pathogenesis and give new insights in the development of new therapies with improved tolerance and efficiency.

## Figures and Tables

**Figure 1 viruses-13-00778-f001:**
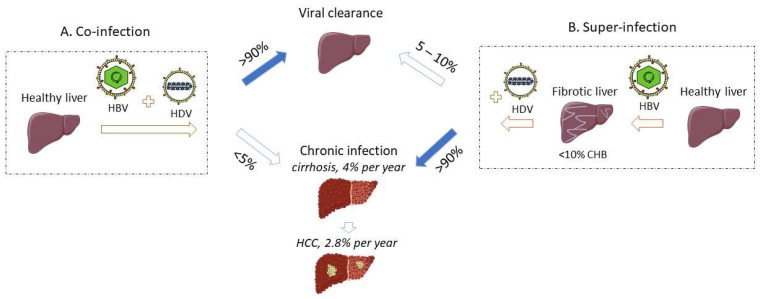
HDV-HBV co-infection or HDV super-infection of a chronically infected HBV patient can progress towards viral clearance or chronic infection. Solid arrows demonstrate the major clinical outcome upon co-or super-infection. Created with Servier Medical ART (SMART).

**Figure 2 viruses-13-00778-f002:**
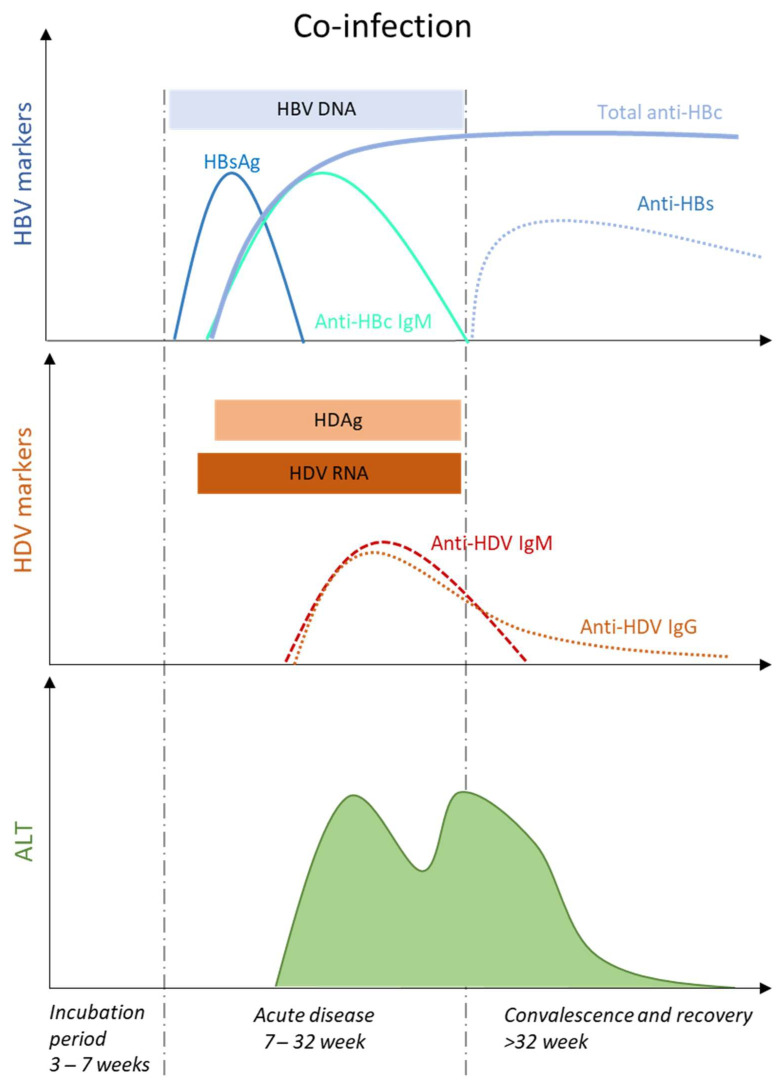
HDV-HBV co-infection will mostly lead to acute hepatitis that is characterised by periods of viral incubation, acute disease, convalescence, and viral resolution. The presence of different serological and biochemical markers characterizes each period. The evolution towards chronic hepatitis is rather rare. Adapted from [15].

**Figure 3 viruses-13-00778-f003:**
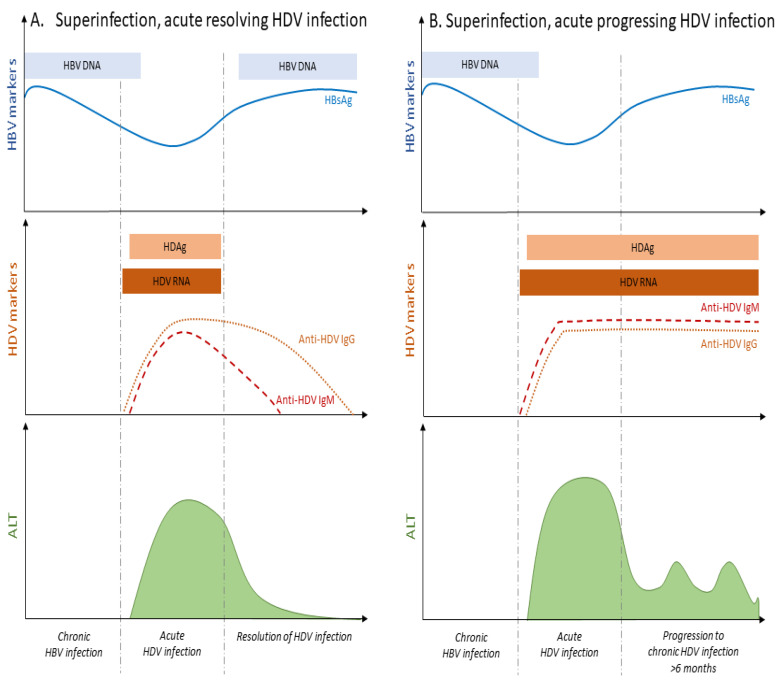
HDV superinfection can evolve to (**A**) acute resolving or (**B**) acute progressing HDV infection in chronically HBV infected patients. The presence of different HBV and HDV serological and liver biochemical markers coincide with a different phase of the viral replication. While a small number of HDV superinfected patients will resolve HDV infection, the vast majority will progress to CHD, the most severe form of viral hepatitis that can lead to end stage liver diseases. Adapted from [15].

**Figure 4 viruses-13-00778-f004:**
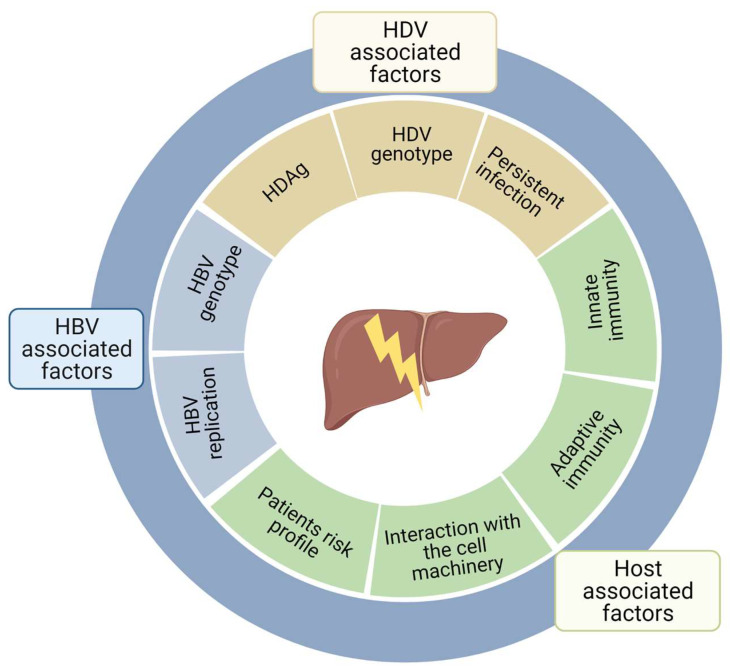
Schematic representation of HDV, HBV and host associated factors that are implicated in HDV pathogenesis.

**Figure 5 viruses-13-00778-f005:**
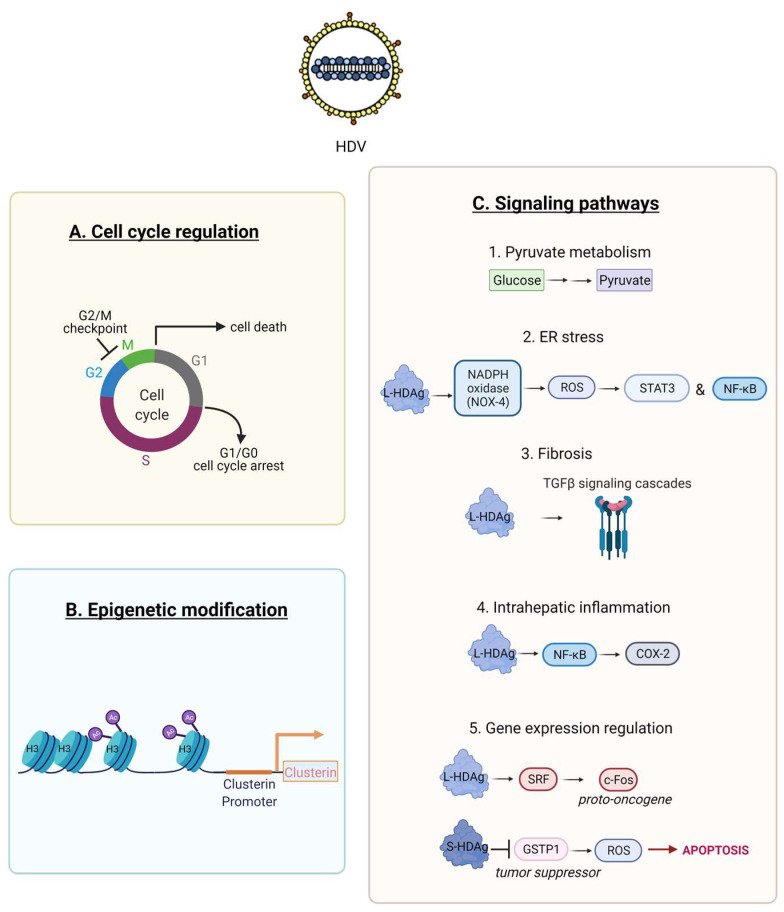
Schematic representation of HDV interactions with the cell machinery factors implicated on HDV induced pathogenesis. HDV can affect cell cycle regulation of the infected cells (**A**), mediate epigenetic modifications (**B**), and activate or downregulate several signalling pathways (**C**). So far, we do not have detailed information for all the implicated interactions of HDV with the cell machinery components. Created with BioRender.com.

**Figure 6 viruses-13-00778-f006:**
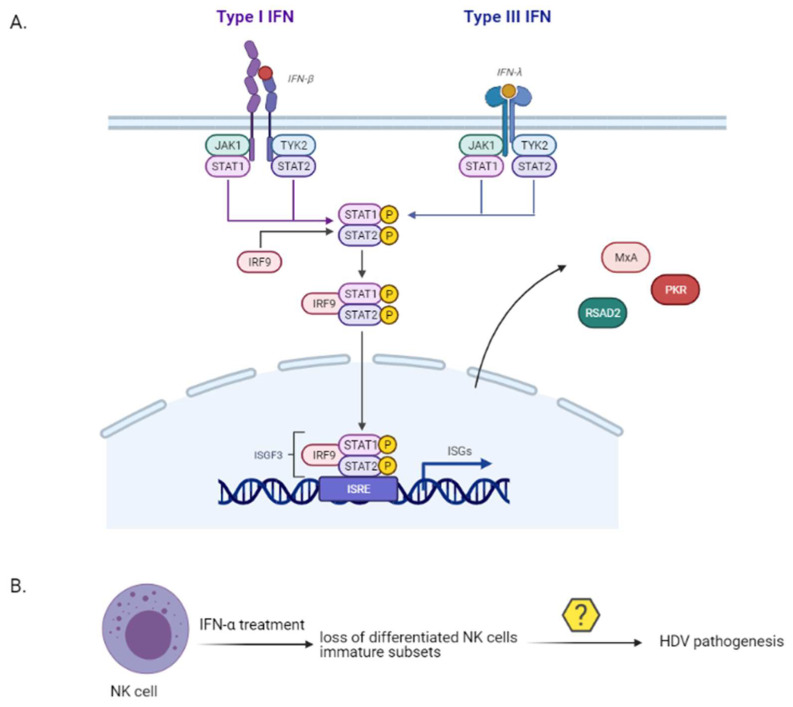
HDV triggers a strong innate immune response in the infected hepatocytes. An unbalanced innate response could contribute to HDV mediated liver immunopathogenesis. (**A**). HDV active replication induces IFN-β and IFN-λ responses in vitro and in vivo and will induce the production of different ISGs. (**B**). IFN-α treatment in chronically infected HDV patients was correlated with a loss of terminally differentiated NK cells and an enrichment in immature NK cell subsets. Their role in HDV pathogenesis is still unclear and further studies are required to better decipher their role in HDV pathogenesis. Created with BioRender.com.

**Figure 7 viruses-13-00778-f007:**
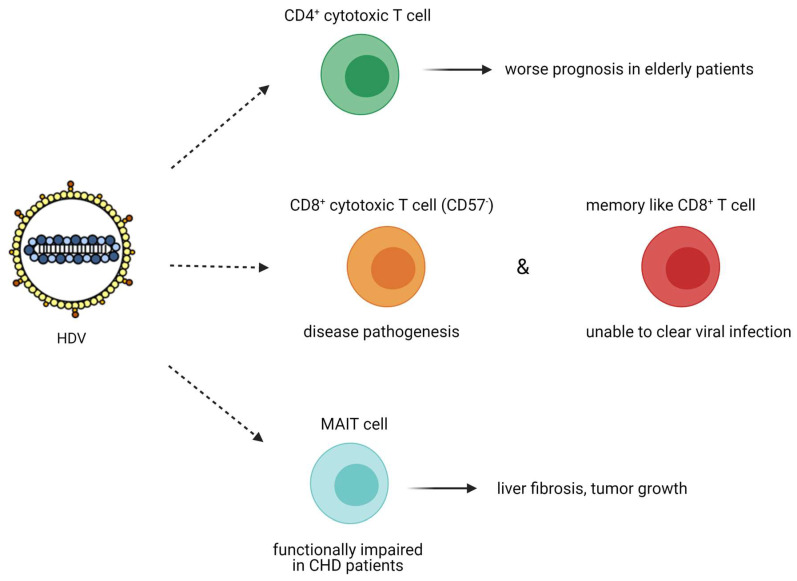
Chronic HDV infection induces an adaptive immune response mediated by CD4^+^ and CD8^+^ cytotoxic T cells and MAIT cells, a subset of innate-like T cells. The increased number of cytotoxic CD4^+^ T cells in patients with advanced liver disease represents a critical factor of severe course of viral hepatitis in elderly individuals. Chronic HDV infection can induce the production of HDV specific CD8^+^ CD57^-^ cytotoxic T cells that contribute to disease progression and memory, such as CD8^+^ T cells, that seem unable to clear viral infection. The role of MAIT cells in HDV pathogenesis needs to be further elucidated, although, it was shown that chronic HDV infection impairs their function. Created with BioRender.com.

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
