# Peer review of "HDV Pathogenesis: Unravelling Ariadne’s Thread"

_viruses, 2021, doi:10.3390/v13050778_

Round 1
Reviewer 1 Report
Comments:
Line 113: HBsAg and HBV-DNA should be added
Line 117: What are the "markers of HDV infection" mentioned here ? Do you mean HDV-RNA? or even HDAg?
Figure 2 : HDAg is drawn, however not mentioned in the text above. What comments the authors can made according to this marker, as it is very rarely found (and used) in HDV diagnosis ?
Line 154: We cannot understand this sentence "Since the HBsAg status is often unknown ... ".
166-167: What do the author mean by "Markers of HDV replication"? The authors should explicit this, as there is to our option only one : the HDV RNA
Line 170: The authors have to be clearer: which "markers of HBV replication" are "usually suppressed and remain undetectable"? We never found in our experience a disappearance of HBsAg for instance.
Line 198: It would be better to write: " active replication of HBV and/or HDV remains ...".
Line 200: How the authors can document this as HDAg is not a suitable and a quantitative tool in HDV diagnosis. Indeed HDAg is quite near found positive in infected patients, mainly due to HDVAb response.
Line 201: How can you quantify "high titers of IgM and IgG antibodies" in clinical practice ?
Line 269: The subtitle should be better: " HDV genome replication in the absence of HBV"
The end of the paragraph 1.3 should be re-written and shorten as the evidence for the transmission of HDV by another helper is so far very tenuous and shown in in vitro experimental models and in a single in vivo study and on one strain.
Line 383: HDV subgenotypes that reflect geographical distribution of HDV strains have been described in a recent paper (Le Gal et al., 2017) and should be mentioned here.
Paragraph 2.1.3 : should also mentioned a recent paper (Roulot et al., 2020) that address the question of HDV-RNA persistance in the severity of the disease.
Line 675: The authors should be more precise here. Indeed the HBV-DNA is very often suppressed but not HBsAg which is also a marker of HBV infection
Reviewer 2 Report
The review article by Tseligka et al. describes in detail the potential host and viral factors implicated in the pathogenesis of acute and chronic HDV-related liver disease, a complex issue due to the satellite nature of HDV. It is un updated and comprehensive report, which touches on several important aspects related to HDV pathogenesis. I have only a few specific comments.
Specific comments
- The initials of the hepatitis viruses are always written in capital letters, but they should be in lower case. The same for tumor necrosis factor alpha on line 471.
- Line 64: “totally asymptomatic”; totally should be removed.
- Figure 1. The rate of chronicity in the setting of superinfection is higher than 80% (usually 90% or more). Also, in Figure 1, the total rate of viral clearance and chronicity does not sum up to 100%. Thus, the numbers should be corrected in the figure and in the text (see below).
- Lines 106-108: The rate of clearance in the setting of HDV superinfection is not 10-15% but remarkably lower; accordingly, the rate of chronicity is higher than 80% as reported in reference 16 [Caredda et al where 20 out of 21 (95%) subjects developed chronic HDV infection following HDV superinfection]. These numbers should be changed, or the authors should include a different reference, if conflicting results have been reported.
- Line 123: “chapter” should be replaced with “section”
- The heading in figure 3A is not clear. The title should be: A. Superinfection, acute resolving HDV infection, since it is a very rare event that acute HDV superinfection resolves; in Figure 3 B the title should be: Superinfection, acute progressing HDV infection because the figure shows also the acute hepatitis phase in the setting of HDV superinfection.
Round 2
Reviewer 1 Report
COMMENTS:
Paragraph 120 to 129 must be re-written. Indeed, It must be mentioned that HDVAg is not a suitable tool in the HDV diagnosis, as very often negative and as no tools are sensitive enough due to immune complexes. In addition, HDV-RNA viral load is a main tool to be used to confirm the diagnosis and the replication status of the infection and not to « replace » the HDAg test.
HDV diagnosis relies mainly (exclusively) upon total HDV AB and when positive on HDV-RNA viral load
Lane 181 : In many cases, HBV rebound is not systematic
Lanes 261-281 : the sentence must be deleted
Author Response
Comments and Suggestions for Authors
Comments:
Paragraph 120 to 129 must be re-written. Indeed, It must be mentioned that HDVAg is not a suitable tool in the HDV diagnosis, as very often negative and as no tools are sensitive enough due to immune complexes. In addition, HDV-RNA viral load is a main tool to be used to confirm the diagnosis and the replication status of the infection and not to « replace » the HDAg test.
HDV diagnosis relies mainly (exclusively) upon total HDV AB and when positive on HDV-RNA viral load
We thank the reviewer for this comment, and we have revised the lines 120-129 accordingly:
HDV diagnosis relies almost exclusively in the detection of total levels of anti-HDV antibodies, and is subsequently confirmed by the detection of HDV RNA by polymerase chain reaction (PCR) [22]. Serum HDV antigen (HDAg) appears early upon HDV-HBV co-infection, but it disappears quickly, therefore requiring repeated testing [20]. Serum HDAg is not a suitable tool for HDV diagnosis as it cannot be directly detected by enzyme immunoassay or radioimmunoassay due to antigen sequestration in immune complexes with high tittered circulating antibodies [21] and requires the application of immunoblot assay under denaturating technique which is difficult to apply for routine detection [21].
Lane 181 : In many cases, HBV rebound is not systematic
We have edited this line by precising that HBV rebound is not always systematic.
Lanes 261-281 : the sentence must be deleted
We ask the reviewer to specify which sentence she/he would like us to remove or edit as to our opinion this part provide important information regarding the impact of HDV on HCC development.
